# Oncolytic Adenoviruses Armed with Co-Stimulatory Molecules for Cancer Treatment

**DOI:** 10.3390/cancers15071947

**Published:** 2023-03-23

**Authors:** Aleksander Gryciuk, Marta Rogalska, Joanna Baran, Lukasz Kuryk, Monika Staniszewska

**Affiliations:** 1Department of Microbiology, Molecular Genetics and Genomics, Centre of Advanced Materials and Technology CEZAMAT, Warsaw University of Technology, 02-822 Warsaw, Poland; 2Department of Virology, National Institute of Public Health NIH-NRI, 00-791 Warsaw, Poland; 3Valo Therapeutics, 00790 Helsinki, Finland

**Keywords:** oncolytic adenovirus, oncolysis, immuno-oncology, co-stimulatory molecules, cancer therapy, immuno-therapy

## Abstract

**Simple Summary:**

Current cancer therapies are insufficient to cure advanced malignancies and cause off-target toxicity. Scientists have focused on developing new, more efficacious, and selective therapies to overcome these issues. This review describes a modern approach based on oncotherapy, a type of cancer treatment that helps the immune system fight cancer. Thus, genetic modifications of adenoviruses to increase the anticancer properties of the therapy (benefits, critical points, and risk assessment) were reviewed. We analyzed clinical trial databases and highlighted significant achievements in adenovirus oncotherapy in recent years.

**Abstract:**

In clinical trials, adenovirus vectors (AdVs) are commonly used platforms for human gene delivery therapy. High genome capacity and flexibility in gene organization make HAdVs suitable for cloning. Recent advancements in molecular techniques have influenced the development of genetically engineered adenovirus vectors showing therapeutic potential. Increased molecular understanding of the benefits and limitations of HAdVs in preclinical research and clinical studies is a crucial point in the engineering of refined oncolytic vectors. This review presents HAdV species (A–G) used in oncotherapy. We describe the adenovirus genome organizations and modifications, the possibilities oncolytic viruses offer, and their current limitations. Ongoing and ended clinical trials based on oncolytic adenoviruses are presented. This review provides a broad overview of the current knowledge of oncolytic therapy. HAdV-based strategies targeting tumors by employing variable immune modifiers or delivering immune stimulatory factors are of great promise in the field of immune oncologyy This approach can change the face of the fight against cancer, supplying the medical tools to defeat tumors more selectively and safely.

## 1. Introduction

Based on WHO Global Health Estimates, the crude death rate per 100,000 population (CDR) decreased over the past 20 years. However, it was observed that the occurrence and mortality of malignant neoplasms increased in this period by almost 9%. Malignant neoplasms are now the second leading cause of death worldwide. Paradoxically, the most common cause of death (cardiovascular diseases) recorded a decrease in CDR [1]. Surgical excision of the tumor and chemo- and radiotherapy are the main methods of fighting cancer [2]. Chemo- and radiotherapy cause systematic toxicity, which is highly invasive for patients. Due to the non-specificity of those therapies, they also impact healthy cells [3]. Even though those therapies are effective in some cases, in others, the quality of a patient’s life is significantly reduced without any therapeutic effect [4].

Human adenovirus vectors (HAdVs) are promising candidates to overcome the drawback of current anticancer therapies. HAdVs are suitable vectors due to their high transgene capacity and flexibility in various modifications [5]. HAdVs have DNA genomes that are easily manipulated without attenuating viral replication and accommodate various transgenes [6]. HAdV clinical safety was evaluated and documented extensively; oncolytic HAdVs are the most frequently used oncolytic viruses in clinical trials, accounting for up to 42% of all trials [7]. Oncorine is the first oncolytic virus approved by the Chinese FDA in 2006 but not in Western countries. The majority of constructs based on human serotypes, such as ONCOS-102, LoAd703, TILT-123, ORCA-010, and CG0070, are under active clinical development [8]. The HAdV vectors are genetically modified to specifically respond to tumor-specific molecular alterations by replication and subsequent cell lysis [9]. This retargeting strategy reduced deleterious side effects and increased the therapeutic index of oncolytic viruses [9,10]. Moreover, HAdVs cause immunogenic cell death, engaging the immune cells to fight cancer [4]. On the other hand, we have to mention the risks of the HAdV therapy, including the presence of antiviral neutralizing antibodies, complement activation, antiviral cytokines, and macrophage predation. These factors may reduce virus replication and promote a premature clearance of oncolytic viruses [6].

In this review, we described all of the adenoviruses species (A–G), especially focusing on the serotypes primarily used in oncotherapy. We aimed to present the current possibilities that oncolytic viruses have to offer. We described adenovirus biology and modifications used to generate better anticancer capabilities. We focused on the most common transgenes used in the HAdV vectors. Finally, we outlined ongoing and ended clinical trials with the presented species.

## 2. Characterization of the Adenovirus Genome

Human adenoviruses (HAdVs) are non-enveloped DNA viruses. Pathogenic human adenoviruses can infect various human cells, leading to respiratory, ocular, enteric, renal, and hepatic diseases [11]. To date, 113 serotypes have been discovered (hadvwg.gmu.edu). Those serotypes are classified into seven “species”, A–G, formerly called subgroups [4]. Species differ by morphology, DNA homology, and oncogenicity [11,12]. Most species use coxsackievirus and adenovirus receptor (CAR) or CAR and silicic acid (SA) to infect cells [12]. However, species B and D utilize the co-stimulatory molecules CD80 and CD86 as cellular attachment receptors [13]. There are conflicting results [14,15,16,17,18,19] on the role of the membrane cofactor protein (MCP) CD46 in facilitating adenovirus 3 attachment to cells.

Adenoviruses contain a linear genome of 34–39 kb and a double-stranded DNA (Figure 1). Both DNA strands are flanked by two inverted terminal repeats (ITR) [20]. The HAdV genes are classified into early genes (E1A, E1B, E2, E3, E4) and late genes (L1–L5). All genes encode approximately 45 proteins [11,21,22]. Early genes are necessary for protein synthesis and replication of the viral genome. Late genes are responsible for virus escape from the cell and synthesis of the capsid structural proteins [20,23]. E1A is the first gene expressed after HAdV enters the host cell. Especially the E1A conserved region 2 (E1A-CR2) plays a critical role, as it interacts with retinoblastoma (Rb), stimulating the cells and viral DNA replication [21]. Two proteins encoded by the E1B gene are apoptosis inhibitor (19K) and inactivator of p53 (55K), respectively [10]. E1B inhibits apoptosis by blocking the function of the p53 tumor-suppressor protein. Adenoviruses without E1A and E1B cannot replicate in normal cells but replicate and cause cytopathic infections in the majority of tumor cells with lack of antioncogenes such as Rb and p53 [21,24]. The E2 gene encodes the DNA polymerases, the single-stranded DNA binding and terminal proteins [10,21]. The E3 gene provides defense mechanisms by inhibiting immune response [20]. Moreover, E3 encodes the adenoviral death protein. This protein induces cell lysis, making HAdV escape possible [10,21]. The last early E4 gene is involved in late viral gene expression and DNA replication. Proteins encoded by the E4 gene also limit the expression of cell proteins, promoting the expression of viral genes [10,11,21]. As previously mentioned, late genes are responsible for the synthesis of structural proteins. Late genes are placed under major late promoters (MLP): L2, L3, and L5 encode penton base, hexon, and fiber proteins, which are the main capsid structural segments. L1 encodes IIIa protein, which builds the inner side of the capsid. The L4 gene is responsible for the VIII protein synthesis, which combines the hexon components to improve capsid stability [10,20].

## 3. Adenovirus Cell Entry, Replication, and Immunogenicity

The fiber and penton base proteins of the HAdVs can bind one or more of the following receptors: CAR, sialic acid, CD46, and desmoglein-2 [25,26]. To activate the immune system, researchers apply cloning techniques to the HAdV vectors with genes responsible for stimulating the immune response, such as TNF-α, IFN-α, IL-2, IL-12, CD40L, OX40L or GM-CSF (Figure 2) [27]. While it was established that the fiber and penton base proteins of many HAdVs interact directly with cellular receptors, there are still few demonstrations showing that hexon protein directly targets receptors. Moreover, the 4-1BB ligand (4-1BBL, CD137L) incorporated in LOAd703 enhanced immunologic memory and expanded natural killer (NK) cells [28]. When viruses enter cells, components such as virus capsid, DNA, and intermediates produced during the replication process act as potent pathogen-associated molecular patterns (PAMPs) [29]. Pattern recognition receptors (PRRs) detect PAMPs and trigger an inflammatory response, including the release of cytokines and chemokines [29]. In general, oncolytic viruses can reverse tumor’s umor immune escape. The virus-induced oncolysis results in apoptosis, necrosis, necroptosis, pyroptosis or autophagy. These types of cell death lead to the release of signal molecules [30] and damage-associated molecules (DAMP) such as heat shock proteins (HSP), HMGB1, intracellular matrix proteins, ATP, DNA, uric acid, and heparin sulphate [29]. HAdVs induce immunogenic cell death (ICD) in infected cells, an essential process for triggering adaptive antitumor immune responses and antitumoral memory. Thus, ICD is essential for the therapeutic efficacy of HAdVs. Adenovirus-mediated ICD is associated with the induction of antitumor immune responses [8]. Deletion of the E1B19K protein, a Bcl-2 homolog, blocks apoptosis and increases viral spread as well as enhances TNF-α-mediated cell death [30]. HAdVs kill infected tumor cells with features of necrosis/necroptosis. Moreover, autophagy [31] is induced, accompanied by the release of high mobility group box-1 (HMGB-1), calreticulin, extracellular ATP, and heat shock protein 70 (Hsp70) [31].

## 4. Oncolytic Adenoviruses—Genetic Modifications to Enhance Oncolytic Activity

Adenoviral therapy is a new approach to cancer treatment. The first oncolytic HAdV vector entered clinical trials in 2001 [12]. The main goal of developing the HAdV oncolytic vectors was to limit their replication in cancer cells. One of the methods to accomplish this was to achieve 24 bp deletion (Δ24) in the E1ACR2 gene, which blocked the ability of E1A to bind to the Rb protein. Consequently, the viruses were not able to replicate in normal cells. However, most cancer cells are defective in the p16-Rb pathway, so the E1A activity is not necessary to unlock the virus replication [5,10,32]. The second strategy to achieve selective replication in cancer cells was to put the E1A gene under a specific promoter, which activated E1A expression only in specific cancer or tissue types [5,12]. The next step in designing the oncolytic viral vectors was to increase their lytic properties. Thus, the deletion of E1B55K or E1B19K in the E1B gene was performed. This prevented the virus from blocking apoptosis in the infected cells, which improved the overall oncolytic potential of vectors [10,21]. Another common change in the oncolytic HAdV genome was the deletion of the E3 region. There were several benefits associated with this modification. Firstly, the lack of the E3 region impaired evasion of the immune system by HAdV. Generally, it is highly required, as targeting the immune system to the cancer cells enhances the anti-tumor response [11]. Deletion of E3 also increased the cargo capacity of the vectors. That made the E3 region a target of transgene insertion [20]. Moreover, the E3 deletion also limited virus replication in cancer cells. A lack of E3 makes the virus vulnerable to the immune system, so vectors are eliminated from the normal cells with active innate antiviral response [33]. While the deletion of E2 and E4 was used in the second generation of non-replicative HAdV vectors, in the oncolytic HAdV vectors, the deletion of those genes was not applied [20]. Since oncolytic viral therapy aims to replicate the virus in the cancer cells selectively, the structure and function of the HAdVs’ core protein must be maintained [10,34]. DNA polymerase, binding protein, and terminal protein (TP) encoded by E2 play important roles in the HAdV life cycle. TP is required for efficient DNA replication initiation, to prevent false internal start, and to protect viral DNA from the nuclease activity [35]. Moreover, the E4 proteins (Orf3 and Orf6) modulate gene expression and replication through interactions with host cell systems [10,34,36]. E4 Orf6/7 modulates the activity of the cellular transcription factor E2F, transactivating the E2 early promoter. E4 Orf4 regulates protein phosphorylation in the infected cells [36]. Furthermore, late genes were modified to obtain a better oncolytic effect. One of the modifications occurred in the L5 gene encoding fiber knob, where genes from different virus species were combined. For example, combining the L5 gene of HAdV-C5 and HAdV-B3 ensured that the virus was able to infect cells with both the CAR receptor (HAdV-C fibers) and the CD-46 or DSG-2 receptor (HAdV-B fibers) [26].

### 4.1. Oncolytic Species B and C and Their Chimer-Versatile Platforms for Cancer Therapy

Species B is less studied than C; thus, most HAdV oncolytic vectors are developed based on HAdV-C5 [37,38]. On the other hand, the HAdV-C5 vectors are limited to cancer cells with the CAR receptors on their cell membrane. The prevalence of species B over C in oncolysis was shown in studies by Mach et al. [39]. HAdV-B3 and HAdV-B35 showed 100-fold higher activity than HAdV-C5 against breast cancer cells. The activity imbalance was due to a lack of CAR receptors on examined cell lines [39]. Thus, trials are underway to use HAdV-B vectors in oncolytic treatments [40,41]. Another study compared the oncolytic activity of the HAdV-B35 modified vector with the HAdV-C5 popular vector. HAdV-B35 was prepared by the combination of pAd35-hTERT-E1 with pAdMS2. The first plasmid contained the E1 gene mutated under the human telomerase reverse transcriptase promoter. The second one included the HAdV-B35 genes with deletion of the E1 gene, ensuring replication defect of the virus. However, the supplier of the mentioned plasmid does not provide deletion description in E1 gene (HepaVec, Berlin, Germany). Results showed higher oncolytic activity of HAdV-B35 in the CAR-negative cancer cells. The oncolytic activity of both HAdV types (B and C) in CAR-positive cancer cells was similar [41]. These results differ from those obtained by Baker et al. [42], where Chinese hamster ovary (CHO) cells with the CAR receptors treated with the HAdV-B35-C5 chimeric vector were not transduced and showed insignificant oncolytic activity. Chimeric vectors are created by merging parts of at least two virus serotypes. In this study, HAdV-B35-C5 had a HAdV-C backbone with additional genes encoding the HAdV-B35 fiber knob. The chimer contained receptors from both species B and C, respectively: CD46 and CAR. An interesting concept was shown by Huang et al. [43], who also created the HAdV-B35-C5 chimer. The chimeric vector infected the SCOV-3 and HO8910 cancer cell lines more efficiently than HAdV-C5. Kuryk et al. [26] worked on the HAdV-B3-C5 chimer with the HAdV-C5 backbone and addition of the HAdV-B3 fiber knob. It allowed the extended options of vector binding not only to CAR but also to the DSG2 and CD46 receptors. It suggests that multispecies chimers can potentially be used to create a universal oncolytic vaccine, independent of the types of receptors on the cancer cell surface.

HAdV-B oncolytic activity is based not only on the virus-lytic cycle of cancer cells but also on the activation of different immune cells. Leung et al. [44] showed that the HAdV-B3-B11p chimer (HAdV-B11 backbone and HAdV-B3 fiber knob) was able to activate NK cells without infecting them. The oncolytic effect of HAdV acting with NK cells was more significant than that of HAdV or NK acting alone. Chen et al. [45] observed high activity of the HAdV-B11 serotype against ovarian cancer in vivo. This activity is associated with the phenotypic change of cancer cells from epithelial to mesenchymal. As the transformation can lead to the formation of cancer stem cells (therefore, to metastasis), the use of HAdV-B11 can be limited.

A common strategy to increase the oncolytic activity of the developed HAdV-B vectors is to add transgene into the wild-type HAdV genome. Zafar et al. [46] showed that adding the CD40L transgene to HAdV-B3 does not affect tumor cell transduction. Expression of CD40L by infected cells increased antitumor response by inducing dendritic cell (DC) maturation, resulting in the activation of T cells both in vivo and ex vivo. In another preclinical trial, Freedman et al. [47] used the HAdV-B chimeric species for fibroblast protein activation and targeted bispecific T cell engager (FAP-BiTE) transgene. The vector with the FAP-BiTE transgene was cytotoxic for both cancer cells and cancer-associated fibroblasts (CAF) in the presence of T cells after infecting the ovarian cancer cells (SCOV-3). In contrast, the non-transformed vector was not toxic for those cells. Correct selection of transgenes extended the activity of the developed HAdV-B vectors to other non-cancer cells, such as the cancer-associated fibroblast. Huang et al. [43] used the HAdV-B35-C5 chimer with the SIRPα-IgG1 Fc fusion gene (SF). It was shown that the product of SF expression recruits macrophages and NK cells, increasing the oncolytic effect of the therapy. The main goal of cloning the transgene into the HAdV-B vector was to increase its oncolytic or transduction properties. However, Bahlmann et al. [48] analyzed how the addition of the junction opening peptide to HAdV-B3 affects its transduction to cancer cells. Results showed a decrease in virus transduction compared to the wild-type HAdV-B3. It is associated with the loss of the ability of modified HAdV-B3 to bind CD46 receptors on the target cells.

The biology of species C human AdV serotype 5 (HAdV-C5) is well-characterized. It is widely used as a gene delivery vector of transgenes in gene therapy and basic science research. It was shown [49] that hexon protein is critical for virus infection of hepatocytes in vivo and human coagulation factor (FX) binding to Ad5. The evolution of species HAdV-C was studied by generating 51 complete genome sequences from circulating strains. Clustering the whole HAdV-C genome sequences confirmed classical typing results (15 HAdV-C1, 30 HAdV-C2, four HAdV-C5, two HAdV-C6). However, two HAdV-C2 strains had a novel penton base sequence and thus were re-labelled as the novel type HAdV-C89. Fiber and early gene region 3 (E3) sequences clustered always with the corresponding prototype sequence, but the clustering of the E4 region indicated recombination events in 26 out of the 51 sequenced specimens. Recombination of the E1 gene region was detected in 16 circulating strains. As early gene region sequences are not considered in the type definition of HAdVs, the evolution of HAdV-C remains on the subtype level. Nonetheless, recombination of the E1 and E4 gene regions may influence the virulence of HAdV-C strains [49].

Another disadvantage of HAdV-C is its common presence in the environment. Thus, humans have fewer anti-HAdV-B than anti-HAdV-C antibodies [50,51]. A number of studies found that poor recognition of HAdV-B by the human immune system prevents vectors from deactivation and makes oncolytic therapy more efficient [4,41,52]. Research on how the immune system reduces the HAdV vector activity was carried out by Di et al. [53]. The authors used custom adenovirus B (ColoAd1, also known as Enadenotucirev), whose genome was a chimer of HAdV-B3 and HAdV-B11. Based on the IC50 values (viral titer causing the death of 50% of cancer cells), the authors pointed out that the custom vector (ColoAd1) displayed 50-fold higher oncolytic activity than HAdV-C5. In 20% (*v*/*v*) human blood serum (with natural human antibodies), the oncolytic activity of HAdV-C dropped 100-fold more than the activity of HAdV-B. What is more, in washed human blood, HAdV-B was not deactivated by blood cells when the activity of HAdV-C was 1000-fold lower [53]. In studies on the same chimer, Dyer et al. [52] showed that the chimer virus kills cancer cells faster than wild-type HAdV-B3, HAdV-B11, and HAdV-C5. The cancer cells were killed by a pro-inflammatory response. The safety of the Colo-Ad1 was first confirmed in phase 1 studies finished in 2016 [54]. The greatest risk of serious complications may occur in patients with immunosuppression. In those cases, antiviral cidofovir can help overcome negative symptoms [55].

Researchers are trying to use nonhuman primate viral vectors to overcome the inhibition of the HAdV vectors by the activity of the immune system. Bots et al. [4] examined HAdV-B isolated from great ape species. The results showed that human immunoglobulins do not neutralize isolated HAdV-B. The nonhuman primate HAdV-B showed more significant oncolytic activity than HAdV-C and nonhuman primate HAdV-C.

### 4.2. Species D and Its Chimer with Species C Adenoviruses in Oncolytic Therapies

Species D adenoviruses are a large, diverse group containing 73 serotypes. It represents more than 70% of all known human adenovirus types [12,56]. The wild-type HAdV-D is associated with ocular infections, but the spectrum of diseases caused by this type of virus is still unclear [11,57]. Infections with HAdV-D occur more often in patients with compromised immune systems, such as those with HIV [42]. Adenoviruses of this species can interact with host cells by CAR receptors (most of HAdV-D), CD46 receptors (e.g., HAdV-D26, -D37, -D56), or salicylic acid (e.g., HAdV-D37) [12,56,57]. Some serotypes of this species can bind to more than one receptor. For instance, HAdV-D17 can interact with the CD46 and CAR receptors, and HAdV-D37 can bind to both CD46 and salicylic acid [58]. Due to HAdV-D having short fibers, their interaction with the CAR receptors is weaker than that of other CAR-specific adenoviruses [57]. HAdV-D is mainly used as a vector in vaccines for infectious diseases. The most popular ones are Ebola and COVID-19 vaccines [11]. Similarly to HAdV-B, species D is less common in the natural environment than HAdV-C. Regarding viral vaccines, this is an advantage, as fewer antivaccine antibodies will be present in the patient’s blood. However, species D adenoviruses are less immunogenic than species C, which may limit the vectors’ application as the vaccine [11]. HAdV-D resistance to seroprevalence was confirmed by vaccinating mice displaying anti-HAdV-C5 immunity with HAdV-D and HAdV-C5 as the references. Results showed that HAdV-D was more immunogenic in the anti-HAdV-C5 mice than HAdV-C5. Nevertheless, in mice without anti-HAdV-C5 antibodies, all tested HAdV-D serotypes (HAdV-D26, HAdV-D48, HAdV-D49) were less immunogenic than HAdV-C5 [51].

Species C serotype 5 is primarily used in oncolytic therapies. However, this type of virus shows low activity against hematologic cancers. Bloodstream cells do not have CAR receptors, which are necessary for HAdV-C to infect cells. Chen et al. [59] conducted extensive research to compare the oncolytic activity of different viral species against B-cell tumors. The result showed that HAdV-D26 and HAdV-D48 had higher oncolytic activity than HAdV-C5 and HAdV-C6. Interestingly, even though HAdV-B does not need CAR receptors to infect cells, this species did not show any significant oncolytic activity against B-cell cancer. Senac et al. [60] showed that HAdV-D26 and HAdV-D48 are oncolytic against myeloma cancer. In this study, HAdV-C5, HAdV-D26, and HAdV-D48 replicated in cells up to 100-fold more than adenovirus species B and F. All tested species (B, C, D, F) were not lytic against normal marrow cells (CD138). Interestingly, while species D did not replicate in normal marrow cells, a 100-fold increase in HAdV-C5 genome copies was noted in the sample from one patient. Based on the results [60], HAdV-D26 and HAdV-D48 were indicated as promising vectors in the anti-myeloma vaccines. Weaver et al. [61] pointed out that HAdVs are powerful tools in oncolytic therapies to fight already growing cancer than in prophylactic anticancer vaccination. In this study, the HAdV vaccine did not protect mice from the beginning of cancer growth. However, HAdVs showed high oncolytic activity against lymphoma in mice, leading to cancer cell death. Data revealed that the oncolytic activity of HAdV-D26 and HAdV-D48 was higher than that of HAdV-C5 [61]. Chen [45] claimed that although HAdV-D overcame some disadvantages of HAdV-C and showed lytic activity against hematologic cancers, this species is not oncolytic for most solid cancers. Recent studies showed that species D has high oncolytic activity against breast cancer. Mach et al. [39] examined the oncolytic activity of many HAdV serotypes from different species against breast cancer. From 10 HAdV-D serotypes, HAdV-D69 and HAdV-D20 showed higher oncolytic activity against the MCF-7 breast cancer cell line compared with HAdV-C5.

Some research focused on creating chimeric HAdV-C-D to combine the advantages of both C and D species. Baker et al. [42] created a chimer from HAdV-C5 and HAdV-D49. This HAdV-C5-D49 chimeric virus (HAdV-C5 backbone and HAdV-D49 fiber knob) could transduce cells independently of the cell receptor type. Chimeric adenovirus was able to transduce both Chinese hamster ovary (CHO) cells with CAR or CD46 receptors. Moreover, the chimer could infect CHO cells without adenovirus receptors, while HAdV-C5 and HAdV-C5-B35 were not active in this cell line. The results showed that other adenovirus receptors such as desmoglein-2, silicic acid, or heparan sulfate proteoglycans were not limiting chimer infection. The transduction of cancer cells by the chimer was 4- to 200-fold higher than that by HAdV-C5 [42]. Developing new chimeric vectors creates universal oncolytic vaccines independent of adenovirus receptors on tumor cells [39]. To date, no clinical trials with oncolytic HAdV-D have been found. Most clinical trials focus on the Ebola and COVID-19 HAdV-D26 vaccines [11,20,62]. We searched the sources: clinicaltrials.gov database, PubChem, ScienceDirect, and Google Scholar.

### 4.3. Species E, F, and G Adenoviruses—Limited in Oncolytic Therapy

To date, serotype 4 is the only one classified as species E. This serotype and, at the same time, the whole species were most likely formed due to the merger of human and chimpanzee adenovirus species (CAdV) [11,63]. In silico analysis showed that the structure of the HAdV-E genome is based on the simian adenovirus CAdV-E26 genome. However, phylogenetic analysis data indicated that in the HAdV-E4 genome, the sequence encoding hexon protein is identical to the sequence in HAdV-B16. That suggests that HAdV-E4 was formed due to recombination of the CAdV-E and HAdV-B genomes [64]. HAdV-E interacts with host cells using the CAR receptor [39], and it is responsible for common respiratory infections in adults [65]. Interestingly, vaccines based on HAdV-E4 were used for 40 years only in the military without application in public health care [66]. HAdV-E4 vectors are currently undergoing clinical trials as vaccines against influenza, anthrax, and HIV [63]. Adenovirus species E is not ordinary in oncolytic therapy. Mach et al. [39] showed that HAdV-E4 displays no transduction into breast cancer cells. Due to the low titer of the virus in tumor cells, oncolytic analysis was not performed on this serotype. Walters [67] performed a comprehensive study to determine the oncolytic activity of different HAdV species. Results showed that HAdV-E4 reduces the viability of cancer cell lines, but its oncolytic activity was lower than that of HAdV-C5 and HAdV-B11.

Species F contains only two serotypes of adenovirus—HAdV-F40 and HAdV-F41. HAdV-A causes gastrointestinal infections [11,12]. Due to their tropism, HAdV-F serotypes have a more stable capsid, ensuring their ability to withstand the harsh conditions in the gut [11]. Adenovirus species F has two types of fibers. The longer fiber enables binding to CAR receptors and transducing the cells. The shorter fiber does not interact with the CAR receptors [12]. Based on the tropism, the oncolytic activity of HAdV-F can be potentially used to fight gastrointestinal cancers [68,69]. One vector based on HAdV-5 was provided with a part (40SA) of the long fiber protein gene of HAdV-F40 [70]. This transgene was reported as a strong splice acceptor. Its presence in front of the hyaluronidase transgene increases the effectiveness of transcription and splicing of that sequence. An increase in hyaluronidase production boosted its activity, which improved the oncolytic properties of the vector [70]. Chen et al. [45,59] excluded HAdV-F species from the broad study of oncolytic vectors due to their inactivity described in earlier studies. Senac et al. [60] used HAdV-F as negative control due to the lack of oncolytic activity. That factor and the low titers of HAdV-F in human cells are reasons why no research focuses on this species.

Only one serotype, HAdV-G52, belongs to the species G [11]. This species has two types of fiber connected to the capsid [25,58,71,72]. Because HAdV-G has two types of fiber, it can interact with the CAR receptors (by the longer fiber) and silicic acid (by the shorter fiber) to enter the cell [11,39,71]. Serotype 52 is associated with intestinal infections and is the least common adenovirus detected in humans [12,39]. The possibility to bind two receptor types and low seroprevalence make HAdV-G52 an attractive candidate as a viral vector [71].

The oncolytic efficacy of G adenovirus was investigated by Mach et al. [39]. HAdV-G52 showed transduction higher than that of HAdV-C5 in three of four tested breast cancer cell lines. Based on these results, HAdV-G52 was proposed as the most effective vector in transgene transformation to cancer cells in this study. Moreover, serotype 52 was the most oncolytic adenovirus against Hs 578T breast cancer cells. Oncolytic activity similar to or slightly higher than that of HAdV-C5 was observed in other tested cell lines. That makes HAdV-G52 a promising vector in breast cancer therapy [39]. To date, no other reports of HAdV-G52 oncolytic activity have been published. Nevertheless, in the near future, some new manuscripts are likely to be published. Liaci [72] suggested that his team will examine the unique properties of HAdV-G52 in future research.

### 4.4. Species A Adenoviruses Excluded from Anticancer Therapy

Adenovirus species A is a small group of four serotypes: HAdV-A12, HAdV-A18, HAdV-A31, and HAdV-A61 [12]. HAdV-A causes gastrointestinal infections [11,68]. HAdV-A transduces cells by interaction with the CAR receptor. This interaction was proved by obtaining the crystal structure of HAdV-A12 and the CAR receptor complex [38].

Chen et al. [45] excluded this species from the studies because of possible carcinogenesis. In vivo studies showed that injection of HAdV-A12 into mouse brain induced cancerous brain tumors [73,74]. Only one study [75] examined the oncolytic activity of HAdV-A. Li et al. [75] concluded that recombinant HAdV-E1A12 targets androgen receptor (AR)-mediated transcription and kills prostate cancer (PC) cells.

## 5. Transgenes Enhancing Efficiency

The natural oncolytic activity of adenovirus can be improved by inserting transgenes into their native DNA [76]. Those transgenes are responsible for modifications in virus structure (capsid, fibers) or secretion of immune-activating peptides [2]. A proper understanding of the implications of various immune-affecting factors for cancer cells is crucial in developing modern anticancer viral therapies. As some of them can act as carcinogenic (TGF-B, IL-10, prostaglandin E) or anticancer (IL-2, IL-12, IL-15) agents, or both (GM-CSF, TNF), examination of new potential transgenes and their combination is necessary to obtain a universal, selective vector. Furthermore, the CD40 ligand (CD40L, CD154) induces apoptosis of tumor cells. It triggers several immune mechanisms, including a T-helper type 1 (T(H)1) response, which leads to the activation of cytotoxic T cells and reduction of immunosuppression [77]. Delolimogene mupadenorepvec (LOAd703) is an oncolytic adenovirus (serotype 5/35) that encodes for the transgenes CD40L and 4-1BBL, which activate both antigen-presenting cells and T cells. Moreover, LOAd703 infects cells via CD46, which enables B-cell infections [78]. Member of the tumor necrosis factor (TNF) superfamily OX40 ligand (OX40L) activates T cells [79]. On the other hand, vectors with immunostimulatory transgenes carry a risk of overdose. Immunostimulatory agents such as recombinant cytokines (INF-α, IL-2) used in the treatment of cancer have adverse consequences, including the acute phase response, cell and tissue abnormalities/injury, cytokine release/cytokine storm, tumor lysis syndrome, vascular leak, and autoimmunity [80]. The concentration of highly toxic cytokines could increase drastically if the vector replicates in patients’ cells with high efficiency. This could lead to serious systemic immune responses in patients [80]. When designing an oncolytic vector with the immunostimulatory transgene, it is crucial to consider the risks involved. This chapter describes the most common immune-effective factors with potential applications in virus oncotherapy.

### 5.1. Granulocyte-Macrophage Colony-Stimulating Factor (GM-CSF)

Granulocyte-macrophage colony-stimulating factor (GM-CSF) is a 23 kDa glycosylated protein classified as a cytokine [81,82]. It acts as a growth and activating factor for myeloid cells (granulocytes and monocytes), macrophages, dendritic cells, CD4^+^, and CD8^+^ T cells [32,81,82,83,84,85]. It is also known to improve the presentation of the antigen by antigen-presenting cells (APC) [86]. When GM-CSF binds to its receptor, various signal cascades are activated (JAK/STAT, PI3K/AKT, and RAS/Raf1/MEK), leading to the transcription of the genes responsible for the proliferation and differentiation of blood cells [82]. This feature can support cancer therapies [21,32]. GM-CSF is used with chemo- and radiotherapy to prevent neutropenia [85]. However, some reports indicate the high toxicity of GM-CSF, which can limit its applications in systemic administration. Secretion of GM-CSF by cancer cells infected with a modified adenovirus vector is a new strategy to overcome GM-CSF toxicity. It provides local protein distribution only in the tumor’s immediate vicinity [32]. The safety of the GM-CSF transgene in the oncolytic vector was confirmed in a clinical trial with the first oncolytic viral vector, which the FDA finally approved. This vector was based on herpesvirus with GM-CSF transgene [87].

Although GM-CSF can support cancer treatment, some reports indicate it can promote cancer growth. In some solid tumors, the hematopoietic process is blocked. In those cases, GM-CSF stimulates myeloid cells to differentiate into myeloid-derived suppressor cells (MDSC). Tumor cells can use those MDSC cells to create an immunosuppressive environment near the cancer [82]. Aliper et al. [85] reviewed studies on GM-CSF stimulating progression and metastases of solid GM-CSF-secreting cancers, also called GM-CSF-addicted cancers. This suggests that using GM-CSF as an adjuvant should be preceded by identifying the tumor subset. Tähtinen et al. [88] also reported the cancer stimulation effect of GM-CSF in an immunosuppressive melanoma mice model. Nevertheless, ongoing clinical trials have confirmed the positive effect of adenoviral vectors with GM-CSF transgene (e.g., Phase I: NCT01437280; Phase II: NCT02143804; Phase II–III: NCT01438112) [62]. ONCOS-102 is an oncolytic adenovirus armed with human GM-CSF and an Ad5/3 chimeric capsid, promoting T-cell infiltration, particularly cytotoxic CD8^+^ T cells [89]. In the case of locally delivered virotherapy, T-Vec was described as a promising anticancer candidate, since administered in combination with immune-checkpoint inhibitors [38].

### 5.2. Interleukin 2 (IL-2)

Interleukin 2 is a small protein of 15.5 kDa, classified to the 4-α-helix cytokine family. IL-2 acts as a growth and activation factor for CD8^+^ and NK cells [90,91,92]. IL-2 receptor consists of three subunits: CD25, CD122, and CD132. The receptor occurs in three versions: low affinity (only CD25), intermediate affinity (CD122 and CD132), and high affinity (all three domains) [93]. CD25 is mainly expressed by Treg cells. Binding to this domain stimulates the immune repression of Treg cells. On the other hand, CD122 and CD132 are expressed mainly by CD4^+^ and CD8^+^ T cells. Binding of IL-2 to this complex leads to the activation and proliferation of CD8^+^ cells [90]. Binding to the trimeric complex also increases T-cell growth. However, CD8^+^ cells express the CD25 subunit only after TCR stimulation [94]. The antitumor activity of IL-2 was proved in a number of studies [90,91,92]. Many reports point out that only a high dose of IL-2 can be applied in anticancer treatment [90,93]. To date, IL-2 is the only cytokine, besides INF, approved by the FDA in single-agent cancer therapy [95]. Administration of low doses of this interleukin inhibits the immune system, which leads to cancer immunosuppression [90]. Although the systematic distribution of high doses of IL-2 has been used in cancer treatment since 1998, this therapy is associated with high toxicity and moderate antitumor activity. Viral delivery of IL-2 is a more promising approach. The usage of viral vectors provides both targeted delivery of interleukin and its local distribution, decreasing systematic toxicity [92]. However, Liu et al. [91] emphasized that even local distribution of IL-2 by viral vectors can cause high toxicity, but the modifications in the IL-2 structure can reduce its toxic effect. IL-2 was modified by the addition of glycosylphosphatidylinositol (GPI) domain and delivered to the cancer cells by the vaccinia virus. Results showed that modification of interleukin resulted in binding IL-15 to the cell membrane. It reduced toxicity without losing the ability of IL-2 to activate the immune system to fight cancer [91]. Due to the high toxicity of IL-2, research has turned toward other, less toxic adjuvants. The most attractive ones are IL-12 and IL-15 [92]. The cytokine-coding oncolytic HAdV TILT-123 (Ad5/3-E2F-d24-hTNF-alpha-IRES-hIL-2), in combination with immune checkpoint inhibitor, promoted antitumor efficacy in tumors that were injected with this HAdV [96].

### 5.3. Interleukin 12 (IL-12)

Interleukin 12 is a heterodimer constructed of IL-12A (p35) and IL-12B (p40) domains [33,81]. IL-12 is a proinflammatory cytokine released by APC, NK, NKT, and activated T cells (CD4^+^ and CD8^+^) [21,33,97]. It increases IFN-γ release, which activates the Th-1 and NKT cells [81,86,92]. Activation of Th-1 cells is associated with the connection between adaptive and innate immune responses. IL-12 can also increase TNFα and GM-CSF release from NK, T, and B cells. IL-12 has an impact on nonimmune cells in a tumor environment [33]. Antitumor activity of IL-12 was proved in preclinical and clinical trials in the early 2020s. In those studies, recombinant IL-12 was administered systemically [33,98,99], which was associated with high toxicity and low efficiency, limiting this method [33,99,100]. Some clinical trials were terminated due to severe adverse events and patient deaths [99]. Therefore, the use of HAdVs with transgene delivers IL-12 directly to the cancer environment [33,99,100] and reduces the overall toxicity of IL-12 [99]. The positive effect of viral therapy with IL-12 transgene in non-replicating viruses was proved over 25 years ago [33]. The most recent studies focus on viruses with replication limited only to cancer cells [99,101]. Wang et al. [101] created modified IL-12 with deletion of N terminated signal peptide in interleukin structure, preventing IL-12 from being released from the cells. The results showed that expressing modified interleukin was associated with a significant reduction in systematic toxicity in animal models.

### 5.4. Interleukin 15 (IL-15)

Interleukin 15 is a small peptide of around 14–15 kDa, classified to the 4-α-helix family, similarly to IL-2 [102,103]. IL-15 is responsible for promoting the proliferation and stimulation of the activity of many immunologic cells: NK, NKT, CD8^+^, and DC. It can also increase macrophage and neutrophil activity [102,104]. The leading role of IL-15 in the immune system is to provide a long-term T-cell response by supporting memory CD8^+^ T cells. The immunostimulant effect of IL-15 found application in cancer treatment therapies. Many reports showed that immunotherapies based on intravascular administration of IL-15 increased survival rates in mouse models [93]. IL-15 does not cause severe capillary leak syndrome, making it safer in oncolytic therapies than IL-2 [92,105]. However, the use of IL-15 is limited due to its short lifetime and biostability [103]. The latest studies have focused on IL-15 agonist. It is a complex of wild-type IL-15 and soluble IL-15R-α. Studies have shown that IL-15 agonist stimulates the immune system more strongly than wild-type cytokine [103]. What is more, agonists significantly increased the half-life of IL-15. In many studies, IL-15 agonists improved cancer degradation compared to wild-type IL-15 [104]. Viral delivery of IL-15 and IL-15 agonists was examined as a potentially more effective and less toxic approach [103,104]. Kowalsky et al. [104] reported that delivering IL-15 agonist by oncolytic vaccinia virus improved cancer reduction. Zhang et al. [102] showed that oncolytic adenovirus with transgene encoding IL-15 enhances the anti-tumor effect of the therapy.

### 5.5. Interferons (INF)

Interferons are classified into three types based on their cellular receptors. INFs are categorized as cytokines, with over 20 members of this family [106]. INF I (e.g., INF-α) and INF II (INF-γ) are responsible for antiviral and antitumoral responses [92,106]. Those INFs are expressed after the recognition of viral nucleic acids by Toll-like receptors [106]. After binding to the receptor, INF activates the JAK-STAT signaling pathway [107]. Different cell types produce different INFs. Dendritic cells produce mainly INF-α, fibroblasts produce INF-β and NK, and cytotoxic T cells and CD4^+^ T cells are sources of INF-γ [92,106]. Type III interferons (INF-λ) have activity similar to that of INF I, but they are expressed only in some epidermal cells [106]. Studies on cancer immunotherapies showed that anticancer activity is related to increased INF concentration. It suggests that INFs play an essential role in cancer treatment [108]. The most common INFs used in cancer therapies are INF-α, INF-β, and INF-γ [95]. Type III INFs are not commonly examined in anticancer studies [107].

The FDA has approved INF-α for single-agent cancer therapy [95]. The interferon showed great activity against hematological malignancies such as hairy-cell leukemia and chronic myelogenous leukemia. However, monotherapies with INF-α are not as efficient as combined therapies with tyrosine kinase inhibitors [109]. INF-α is also associated with defending tumor resistance. Studies showed that this type of interferon could inhibit the proliferation of MDSC and restrict the production of chemokine CCL2 by cancer cells, which attracts Treg into the tumor microenvironment (TME). Those properties of INF-α are suitable for anti-cancer adjuvant therapy [106].

INF-β is another cytokine that influences cancer development. A recent study by Du et al. [107] concluded that INF-β induces the anticancer activity of HMC. Interestingly, INF-γ did not show improvement in examined therapy. Other studies showed that tumor cells with blocked INF-β production showed higher vascularization [106]. Yoshimura et al. [110] reported that INF-β increased the anticancer effect in combined therapy with tumor necrosis factor-related apoptosis-inducing ligand (TRAIL) compared to TRAIL monotherapy. Blaauboer et al. [111] showed that combining gemcitabine therapy with INF-β gives better results in vitro than monotherapy. However, INF-β loses its ability to promote the immunologic response of the cells after combining with gemcitabine.

INF-γ is the only type II INF expressed by macrophages, NK, and T cells. Production of this interferon is stimulated, among others, by IL-12 and IL-15 [106]. INF-γ is considered a connector between adaptive and innate immune responses. Low doses of INF-γ lead to tumor metastasis [112]. Even though there are some reports of a potential cancer genesis effect of INF-γ, the vast majority of studies have confirmed its anticancer properties [112,113]. Modern approaches use INF-γ as an adjuvant in cancer therapies. Peng et al. [114] showed that combining 5-fluorouracil therapy with INF-γ leads to an increase in thymidine phosphorylase expression, which reduces cancer growth rates. Compared to monotherapy, the combination with INF-γ improved tumor inhibition by almost 60% [114]. INF-γ is also known to reduce angiogenesis by inhibiting vascular endothelial growth factor (VEGF) [113].

Although INFs have antiviral activity, some trials were conducted to incorporate gene-encoding INFs in oncolytic viral vectors. Reports showed that expressing INFs can improve the overall anti-cancer effect of viral therapy. This result was accurate for all types of INF, including INF-α, INF-β, and INF-γ [106]. In another report, an oncolytic virus with INF-γ transgene activated DC more efficiently than wild-type OV. As a result, the tumor growth was reduced [115]. Salzwedel et al. [116] proved that INF encoding oncolytic virus radically improves chemoradiotherapy. A study showed that the antiviral activity of INF-β can also be used in viral therapy. Expressing this cytokine limited virus replication in normal cells, targeting the therapy in cancer cells [117].

Even though INF exhibits anticancer activity, its usage in therapies is limited due to its high toxicity. What is more, in some cases, INF can stimulate cancer genesis and metastasis [112]. INF-γ and INF-β can also induce PD-1L production in some cancer cells. That leads to increased immunosuppression of TME, which is negative in cancer therapies [106,112]. Taking this into account, caution should be exercised when designing INF-based therapies.

### 5.6. Tumor Necrosis Factor (TNFα)

Tumor necrosis factor is a homotrimeric peptide with 17.5 kDa monomers. TNF-α can bind to two types of receptors: TNFR1 and TNFR2. The first one is the death receptor, which most human cells express. The second is expressed only by immune cells and does not contain a death domain [3,113]. TNF anticancer activity has been proved in many studies over the years [118]. Tähtinen et al. [88] used TNF-α in adoptive T-cell therapy of a highly immunosuppressive melanoma mouse model. Intratumoral administration of TNF resulted in a significant reduction in tumor cells in mice. The antitumor potential of TNF was caused by both the antitumor activity of TNF and an immunostimulant effect engaging CD8^+^ T cells. In another study, Xia et al. [118] reported that liposomes containing TNF could significantly boost PD-1/PD-1L anticancer therapy. The reports [88,113] claim that TNF can also reduce tumor vasculature, which is another mechanism for reducing cancer growth. However, tumor growth inhibition occurs only when TNF concentration is rapidly increased. In small concentrations, TNF acts as a tumor stimulator, increasing reactive oxygen species and promoting DNA damage and mutations [92,119]. Systemic administration of TNF was associated with high toxicity, which limited this therapy [3,120]. The effective method of direct distribution of TNF involves oncolytic viruses with a TNF transgene [118]. Viral vectors offer the opportunity to provide high concentrations of TNF with a low systemic release, decreasing the therapy’s toxicity. The advantage of adenovirus-based gene therapy is the ability to target TNF directly in cancer cells by limiting viral replication in healthy cells [88]. Cervera-Carrascon et al. [121] showed that treatment with adenoviral TNF/IL-2 vector combined with anti-PD1 antibody results in 100% recovery in mouse models with melanoma tumors.

## 6. Oncolytic Adenoviruses: Clinical Progress

The clinical trials with oncolytic HAdV are presented in Table 1. To our knowledge, clinical trials in virotherapy have mainly used species B and C to date. Most clinical trials have focused on the HAdV-C5 vaccine vectors; however, some interesting trials have examined species B.

Hemminki et al. [40] performed preclinical and clinical trials with the modified HAdV-B3 virus. Its E1A promoter was replaced with human telomerase promotor (hTERT), preventing the virus from replication in normal cells and intensifying its replication in cancer cells. The studies were conducted on a group of patients with chemotherapy-resistant cancers. Preclinical trials showed that the created HAdV-B3-hTERT-E1A vector had the same or higher oncolytic activity in vivo compared to HAdV-C5 or the HAdV-C5-B3 chimer. Clinical studies found that 4 × 10^12^ VP does not correlate with adverse reactions. In more than 60% of patients, the vaccine showed antitumor activity, but the analysis was hampered due to the swelling of the tumor [40].

According to the clinicaltrials.gov database, a phase I clinical trial is currently underway, and four trials were completed on Enadenotucirev (EnAdV or ColoAd1) as a chimer of HAdV-B11 and HAdV-B3 [62]. The first trial started in 2012. The study included both phase I and II clinical trials. Phase I focused on determining the maximum tolerated dose (MTD), and the vaccine’s safety was confirmed up to 3 × 10^12^ VP [122]. Unfortunately, data from phase II were not provided. Another phase I trial, including Enadenotucirev, started in 2013; the results showed a high distribution of viruses in over 80% of cancer cells, and the vector replication occurred only in cancer cells. Moreover, increased stimulation of CD8^+^ lymphocyte proliferation was noted. Based on this observation, both oncolytic and immune-stimulative mechanisms of action were considered [54]. An increase in CD8^+^ infiltration was also reported in another phase I trial performed in 2014. The MTD was established at 1 × 10^12^ VP, which is lower than the results of Machiels et al. [123]. In 2016, a new phase I study started. EnAdV was used in combination therapy with the PD-1 inhibitor nivolumab. The primary outcome was to determine the safety of the therapy and MTD, which is still not described [62]. The subsequent clinical trial started in 2019 and is currently underway (the status is: recruiting). The trial aimed to determine the optimal dose of the vaccine in the range from 1 × 10^12^ VP to 3 × 10^12^ VP. The therapy combines virus oncolytic activity and chemoradiotherapy (Capecitabine and radiotherapy) [62,124]. In 2019, the modified HAdV-B3-B11 chimer was found to be able to express an anti-CD40 antibody, which was tested in another clinical trial. In a similar trial started in 2020, the same chimeric vector was modified to express the FAP-TAc antibody, CXCL9, CXCL10, and IFN-α. Both trials aimed to determine dose escalation and safety profile [31,62].

Only one clinical trial of adenovirus B, other than EnAdV and HAdV-B3, was found in the database. In the study from 2015, a combination of HAdV-B68 with tremelimumab and plasmid DNA was examined to determine clinically significant adverse events (DLTs). This trial was terminated in 2021 due to a strategic reevaluation [62]. No trials over phase II were found.

Anticancer vaccines based on adenovirus species C are more often included in clinical trials than those based on adenovirus B. VCN-01 is the most examined vector, with 14 clinical trials conducted. The vector was modified with a transgene encoding the integrin-binding RGDK motif, inserted in the fiber domain, and a sequence encoding human recombinant hyaluronidase. Deletion in the E1A gene ensured selectivity of replication in cancer cells only [125]. CG0070 is also an interesting vector that entered phase III clinical trials in in 2022. The HAdV-C5 virus carries a transgene encoding GM-CSF. Results from the ended trials proved the safety of this vaccine [126]. DNX2041 and DNX2440 are other examples of vectors with the 24 bp deletion in E1A. To make them more specific against glioma cancer cells, both had RGB encoding sequences inserted in the fiber knob region. DNX2440 also has OX40 transgene, which differentiates this vector from an earlier version (DNX2041) [127,128]. A replication-competent MSC-DNX-2401 demonstrated a favorable safety profile and prolonged survival of patients with recurrent high-grade glioma [62,129].

ONCOS-102 is a vector with 24 bp deletion and insertion of GM-CSF transgene [89,130]. Clinical trials proved that expressing GM-CSF impacts the activation of the patient’s immune system. However, upregulation of PD-1L was observed in the experimental group. It was proposed to associate virotherapy with PD-1 inhibitors [131]. Two transgenes encoding TMZ-CD40L and 4-1BBL were inserted into HAdV-C5 with 24 bp deletion from a LOAd703 vector. Clinical trials proved its safety in combination with chemotherapy. At the highest dose level, ORR was over 55%, confirming the therapy’s effectiveness [132]. ICOVIR is a modified version of DNX2041. Its E1A promoter was replaced with a human E2F-1 promoter. To increase E1A mRNA transcription, the Kozak sequence was inserted into this gene. The safety of the vector was confirmed in phase I trials [133]. ORCA-010 is also a modified HAdV-C5, carrying RGB transgene and 24 bp deletion. The vector has an additional mutation in the E3 gene in the sequence encoding 19K protein [134]. Phase I clinical trials showed a good safety profile. The therapy resulted in significant prostate cancer reduction [135]. Ad5CMV-p53 gene was studied in treating patients with unresectable hepatocellular carcinoma (HCC) or highly suspicious for HCC based on CT scan and elevated alfa-fetoprotein (in phase I) [62]. Randomized trials with Ad5-yCD/mutTKSR39rep-ADP were conducted to assess the safety and efficacy of combining oncolytic adenovirus-mediated cytotoxic gene therapy (OAMCGT) with intensity-modulated radiation therapy (IMRT) in patients with intermediate-risk prostate cancer [62,97]. In the phase I/II, the safety of immunization with HAdV5 encoding the tumor carcinoembryonic antigen (CEA) (Ad5 [E1-, E2B-]-CEA(6D)), in patients with advanced or metastatic CEA-expressing malignancies, was evaluated [62,136]. Telomelysin (OBP-301) gene-modified oncolytic adenovirus, selectively replicated in cancer cells by introducing human telomerase reverse transcriptase (hTERT) promotor, was designed to infect and destroy tumor cells [62]. In phase I trial studies, the side effects of OBP-301 were studied when it was given together with carboplatin, paclitaxel, and radiation therapy in treating patients with esophageal or gastroesophageal cancer [62].

## 7. Limitations of Adenoviruses in Clinical Trials

Oncolytic HAdVs display therapeutic efficacy but also limitations when applied as a monotherapy [31]. HAdVs show outstanding potential to immunoactivate tumors that are unresponsive to systemic immunotherapies. HAdVs are highly versatile platforms for the local delivery of immune-activating factors to modulate intratumoral immune cell contexture and to break immune suppression. The limitations of HAdVs’ efficiency in clinical trials were due to biological barriers, tumor heterogeneity, and immunosuppressive TME [137]. Locally administered oncolytic AdVs have demonstrated promising results. However, their therapeutic efficacy is not yet optimal due to weak intratumoral virion retention, virus shedding to normal healthy organs, variable infection, and replication efficacy due to the heterogeneity of cancer [138]. Therefore, the development of new and more potent oncolytic adenoviruses is in high demand [139,140].

## 8. Commercial Companies Developing Virology-Based Technologies

A range of novel oncolytic viruses that were evaluated for mechanisms-of-action (MOA) and proof-of-concept (POC) has been developed by biotech and big pharma companies. The companies collaborated with research institutes and medical centers to develop and assess oncolytic virus platforms. The companies meet WHO requirements for transparency and publication; they are registered in either a Primary Registry in the WHO Registry Network or an ICMJE-approved registry. The companies are focused on the development of HAdV therapies for different types of cancer and in combination with other cancer therapeutics including chemotherapy and checkpoint blockade antibody therapy. The companies offer to start from nonclinical tests and verification through manufacturing, design, and evaluation of clinical trials to commercialization. Lokon Pharma AB conducts clinical trials (phase I–III) including patients with various types of cancers: pancreatic, ovarian, and colorectal cancers and malignant melanoma. Creative Biolabs^®^ OncoVirapy^TM^ offers the development of HAdVs, their purification, and characterization (www.creative-biolabs.com, accessed on 4 March 2023). Creative Biolabs is active in the field of vaccine development and is focused on cancer vaccines. Batavia Biosciences scales viral vaccine production up and performs advanced product analysis with GMP standards (www.bataviabiosciences.com, accessed on 4 March 2023). Vigene Biosciences (a part of Charles River) and Genscript support the products’ legalization in the pharmaceutical market (www.criver.com, accessed on 4 March 2023, www.genscript.com, accessed on 4 March 2023). Oncolys BioPharma aims to bring therapeutics to early clinical proof-of-concept and expands pipelines through strategic collaborations.

## 9. Conclusions

Oncolytic HAdVs show a safety profile in cancer immunotherapy. Current HAdVs demonstrate therapeutic efficacy but also limitations when applied in monotherapy. HAdVs are versatile platforms for the local delivery of immune-activating factors to modulate intratumoral immune cells. Genetic engineering of virus functions shows high potential for the development of innovative anticancer drugs. Additionally, less common HAdV species have great potential in oncolytic therapy. In particular, adenovirus species B is an interesting candidate for the development of new oncolytic adenovirus vaccines. Preclinical studies showed the high oncolytic activity of this type of vaccine. The advantages of this type of virus can help overcome some HAdV-C limitations. However, creating chimeric HAdV-C5 and HAdV-B3 seems more widespread. It will be with foundational understanding that oncolytic immunotherapies and their delivery will be refined to broaden future horizons in the direct modulation of the tumor microenvironment and, most importantly, to improve cancer therapy.

## Figures and Tables

**Figure 1 cancers-15-01947-f001:**
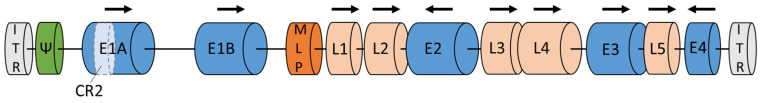
Adenoviral genome. Legend: E1–E4—early genes; L1–L5—late genes; ITR—inverted terminal repeat; MLP—major late promoter; Ψ—packaging sequences. Based on GenBank: MF502426.1; NCBI Reference Sequence: NC_001405.1 and [10].

**Figure 2 cancers-15-01947-f002:**
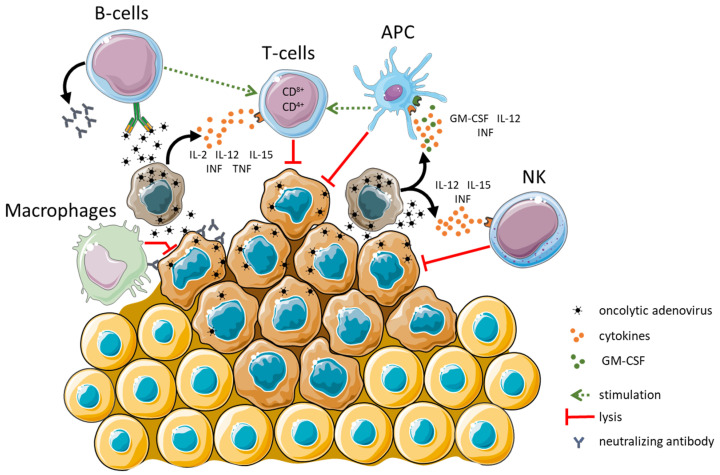
Activation of immune system by oncolytic adenovirus. Drawn by using pictures from Servier Medical Art, licensed under a Creative Commons Attribution 3.0 Unported License (https://creativecommons.org/licenses/by/3.0/, accessed on 21 March 2023).

**Table 1 cancers-15-01947-t001:** Oncolytic adenovirus clinical trials.

Species	Virus Name	Virus Type	Phase	Posted	Clinical Endpoint	NCT Number
B	AdV-B3-hTERT-E1A	HAdV-B3	1	2012	MTD	-
	ColoAd1; EnAdV; Enadenotucirev	HAdV-B11-B3	1/2	2012	MTD	NCT02028442
			1	2013	Distribution	NCT02053220
			1	2014	MTD	NCT02028117
			1	2016	MTD	NCT02636036
			1	2019	MTD	NCT03916510
	NG-350A	HAdV-B11-B3	1	2019	MTD	NCT03852511
	NG-641	HAdV-B11-B3	1	2020	MTD	NCT04053283
			1	2021	AE	NCT05043714
	VBIR-1; PrCa VBIR	HAdV-B68	1	2015	DLT	NCT02616185
C	CG0070	HAdV-C5-E2F	1	2005	MTD	NCT00109655
			2/3	2011	CR, DCR	NCT01438112
			2	2014	DCR	NCT02143804
			2	2015	DCR	NCT02365818
			2	2020	DOR	NCT04387461
			3	2020	DOR	NCT04452591
			1	2020	EAE	NCT04610671
	DNX2401	HAdV-C5-Δ24-RGB	1	2008	MTD	NCT00805376
			1	2013	DLT	NCT01956734
			1	2014	ORR	NCT02197169
			2	2016	ORR, OS	NCT02798406
			1	2017	OS, MTD	NCT03178032
			1	2019	MTD, AE, TR, TP, VS, IAA	NCT03896568
	DNX2440	HAdV-C5-Δ24-RGB	1	2018	DLT	NCT03714334
			1	2021	MTD	NCT04714983
	LOAd703	HAdV-C5-Δ24	1/2	2016	DLT	NCT02705196
			1/2	2017	ST	NCT03225989
			1/2	2018	ORR, OS	NCT03555149
			1/2	2019	ORR, OS	NCT04123470
	ONCOS-102; CGTG-102	HAdV-C5-Δ24-GMCSF	1	2011	ST	NCT01437280
			1	2012	AE	NCT01598129
			1/2	2016	TEAE, DLT	NCT02963831
			1	2016	TEAE	NCT03003676
			1/2	2016	TEAE	NCT02879669
			1/2	2018	OS, ST	NCT03514836
			2	2022	TEAE, ORR	NCT05561491
	VCN-01	HAdV-C5	1	2014	AE	NCT02045602
			1	2014	AE	NCT02045589
			-	2017	TEAE	NCT03284268
			1	2019	AE, ORR	NCT03799744
			1	2021	ORR, DOR, OS, PFS	NCT05057715
			-	2023	TEAE	NCT03284268
	ICOVIR	HAdV-C5	1	2013	MTD	NCT01864759
			1/2	2013	AE	NCT01844661
			1/2	2021	DLT	NCT04758533
			1/2	2021	OS, ORR	NCT05047276
	ORCA-010	HAdV-C5	1/2	2019	ST	NCT04097002
	H101; Oncorine	HAdV-C5	3	2018	OS, PFS, AE	NCT03780049
			2	2021	AE, TTRP	NCT04771676
			4	2021	ORR, DCR	NCT05113290
			4	2021	PFS, ORR, DCR	NCT05124002
			2	2022	CR	NCT05564897
			2	2022	ORR, OS, DCR, PFS	NCT05234905
			1	2023	MTD, DLT, AE	NCT05675462
C	Ad5yCD/mutTKSR39rep-ADP	HAdV-C5	1	2006	TC	NCT00415454
			2	2007	MTD, OS, TC, FFF, CTL, AE	NCT00583492
			1	2016	AE	NCT02894944
			1	2017	MTD, OS, TC	NCT03029871
			2	2021	FFF, ORR	NCT04739046
			1	2023	MTD	NCT05686798
	Ad5-yCD/mutTKSR39rep-hIL12	HAdV-C5	1	2015	MTD, FFF, OS	NCT02555397
			1	2017	TC, AE	NCT03281382
	Ad5.SSTR/TK.RGD	HAdV-C5	1	2009	TC	NCT00964756
	AdHER2.1	HAdV-C5	1	2005	TC	NCT00197522
			1	2006	TC, MTD	NCT00307229
	ADV-hIL-12	HAdV-C5	1	2005	MCL	NCT00110526
			1	2006	TC	NCT00301106
			1	2009	MTD, TC	NCT00849459
	Ad-RTS-hIL-12; INXN-2001	HAdV-C5	1/2	2011	ST	NCT01397708
			2	2012	ST, ORR, CR	NCT01703754
			1	2014	ST, MTD	NCT02026271
			1/2	2015	ST, ORR, CR	NCT02423902
			1/2	2017	ST	NCT03330197
			1	2018	ST, ORR, PFS, OS	NCT03636477
			1	2018	ST, ORR, PFS, OS	NCT03679754
			2	2019	ST, PFS, OS	NCT04006119
C	OBP-301; Telomelysin; Ad5-SGE-REIC/Dkk3	HAdV-C5	1	2014	ST, MTD, DLT	NCT02293850
			1	2017	DLT, RR, PFS, AE	NCT03172819
			2	2017	ORR, PFS, OS, CR	NCT03190824
			1	2017	DLT, AE	NCT03213054
			2	2019	ORR, DOR, OS, PFS	NCT03921021
			1	2020	DLT, AE, CR	NCT04391049
			2	2020	ORR, TC, OS, PFS, DOR	NCT04685499
			1	2020	CR, PFS, OS	NCT04391049
	TILT-123	HAdV-C5	1	2020	ST, AE	NCT04217473
			1	2021	AE	NCT04695327
			1	2022	AE	NCT05222932
			1	2022	AE	NCT05271318
	MG1MA3	HAdV-C5	1/2	2014	AE, ORR,	NCT02285816
	AdcuCD40L	HAdV-C5	1	2006	TC, AE	NCT00328887
			1/2	2007	PE	NCT00504322
	AdCD40L; Ad-ISF35	HAdV-C5	1	2008	TC, ST	NCT00772486
			1	2008	TC, ST	NCT00783588
			1	2008	TC, ST	NCT00779883
			1	2009	MTD, ST	NCT00850057
			1	2009	ST	NCT00849524
			2	2009	ORR	NCT00942409
			1/2	2011	AE	NCT01455259
			1/2	2016	MTD, ORR	NCT02719015
	Ad5-hGCC-PADRE	HAdV-C5	1	2013	AE	NCT01972737
	Ad-E6E7	HAdV-C5	1	2018	ST, MTD	NCT03618953
	ETBX-071—PSA ETBX-061—mucin1 ETBX-051—brachyury ETBX-011—CEA	HAdV-C5	1	2018	DLT, CR, ORR	NCT03481816
	ETBX-061—mucin1 ETBX-051—brachyury ETBX-011—CEA	HAdV-C5	1	2017	AE, CR,	NCT03384316
	VB-111	HAdV-C5	1	2019	ST, PFS, OS	NCT04166383
	AdAPT-001	HAdV-C5	1/2	2020	MTD, DLT	NCT04673942
	RSV-TK	HAdV-C5	1	2004	-	NCT00005057
	Ad-hCMV-TK Ad-hCMV-Flt3L	HAdV-C5	1	2013	MTD, OS	NCT01811992
	Ad-TD-nsIL12	HAdV-C5	1	2023	ST, OS,	NCT05717712
			1	2023	ST, OS,	NCT05717699
	Ad5CMV-p53	HAdV-C5	1	2004	DLT	NCT00003147
	Ad5 [E1-, E2B-]-CEA(6D) ETBX-011	HAdV-C5	1/2	2010	SF, TC, OS, CMI	NCT01147965

Legend: AE—adverse events, CMI—cell mediated immune response, CR—complete response proportion, CS—complete response, CTL—cytotoxic T lymphocyte response, DCR—durable complete response proportion, DLT—dose limiting toxicities, DOR—duration of response, FFF—freedom from biochemical/clinical failure, HAdV—human adenovirus vector, IAA—immunogenicity based on AdV antibodies, MCL—maximum cytokine level, MTD—maximum tolerable dose, ORR—objective response rate, OS—overall survival, PE—product expression, PFS—progression-free survival, RR—response rate, ST—safety, TC—toxicity, TEAE—treatment-emergent adverse events, TP—time to progression, TR—tumor response, TTRP—time to repeat paracentesis, VS —virus replication. Based on: clinicaltrials.gov.

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
