# Peer review of "Oncolytic Adenoviruses Armed with Co-Stimulatory Molecules for Cancer Treatment"

_cancers, 2023, doi:10.3390/cancers15071947_

Round 1

Reviewer 1 Report

In this manuscript Gryciuk and co workers  revise  the AdVs species (A-G) used  as oncolytic viruses and the strategies employed to arm oncolytic adenoviruses. Authors also discuss the results of some clinical trails performed with oncolytic adenoviruses.

The study is interesting and well organised and deserves publication on Cancers

Minor

Line 79  Conserved not converse

Line203    please correct: ..,

Line 265  Dot lacking

Line 436 IL-15 not IL-2  

Line 469 Severe not serve  

Author Response

Dear Reviewer, 

Thank you for the comments. All Authors contributed to making the manuscript better according to your comments. Thank you for the opportunity to revise it. 

Monika Staniszewska 

Reviewer 2 Report

The periodical reviews reflecting some popular directions of R&D are very useful for experimental scientists who have only a limited time to screen the huge amount of publications. But these reviews should be comprehensive and representative reflecting all significant publications in the area of these R&D. The manuscript I have read (“Oncolytic adenoviruses armed with co-stimulatory molecules for cancer treatment”)  is a good and useful review with a very significant list of references. In my opinion, it deserves publication in your journal but the authors should correct some significant issues in it.

1.   The first sentence of Abstract is “Adenovirus vectors (AdVs) are currently the most widely used platforms in clinical trials for human gene delivery therapy”. It is not a correct statement because in the website CLINICALTRIALS.GOV the search by words GENE THERAPY gives 778 references, the search by words GENE THERAPY ADENOVIRUS gives 57 references and search by GENE THERAPY  ADENOASSOCIATED VIRUS gives 95 references. And search by GENE THERAPY AAV – 113 references.

2.   The table with the listed clinical trials therefore does not look comprehensive.  

3.   It would be very useful to include into this review the short description of possibilities offered by commercial companies such as Creative Biolabs, for example.

In my opinion, the suggested corrections will significantly improve the manuscript.

Author Response

(The authors gave the same response as above.)

Reviewer 3 Report

The manuscript by Gryciuk and colleagues reviews the use of human adenoviruses as therapeutic agents in oncolytic virus strategies. This encompasses the range of genetic modifications that are applied, the diversity of transgenes used and intended  to boost anticancer efficacy as well as the various types and species of adenoviruses that are used. As such the contents of thee review spans a wider area than suggested by the title of the manuscript.

The manuscript is well written although some typo’s creeped into the text.

There are several aspects on which the manuscript need some attention.

1) The nomenclature and taxonomy of the adenoviruses needs to be brought into accordance with ICTV guidelines. The adenoviruses described here are all human adenoviruses and should be abbreviated as ‘HAdV’, divided in species HAdV-A to HAdV-G. This is important because there is also adenoviruses isolated from non-human primates (SAdV) also grouped in species SAdV-A to SAdV-J. As a consequence the term ‘AdV-B’ is confusing. Please correct the text of the Abstract and throughout the remainder of the manuscript.

L2) Line 68: to date 113 types have been discovered (http://hadvwg.gmu.edu/)

33)      Line 52-54: adenoviruses can be deleterious to healthy tissues in humans, e.g. in immune compromised patients. This suggests non-cancerous cells can replicate the virus to high levels. Please correct.

44)      Line 54: ref 4 does not state that adenoviruses cause immunogenic cell death. Maybe ref 21 is better here?

55)      The statement that:  HAdV-B and D are unique since they use CD48, CD80 and CD86 as receptors is incomplete and partly erroneous. CD48 should read CD46, and in addition to CD80 and CD86 more receptors can be used, as is correctly outlined below the text in this manuscript. Please correct.

66)      Line 73: The range of genome sizes seems to refer to adenoviruses of other host species, too. For HAdVs a range of 34 -39 kB seems more appropriate.

77)      Line 79: please amend ‘converse’ to read ‘conserved’.

88)      Line 82-83:  The sentence: ‘E1B-55K is also associated with AdV survival in cancer cells [12].’ needs some explanation.

99)      Line 83: please amend ‘DNA binding protein’ to read ‘single-stranded DNA binding protein’.

110)   Line 91: please amend ‘penton’ to read ‘penton base’.

111)   Line 102: Please amend ‘The fibre proteins of AdVs bind ‘ to read ‘The fibre proteins of the various HAdVs can bind one or more of the following proteins as their receptor ‘.

112)   Line 157: It is unclear what part of the transcription machinery is encoded by the adenovirus E2 or E4 genes. Please provide some detail here.

113)   Line 174: please provide some detail of the E1 deletion that was used.  

114)   Line 179-180: please mention what parts of which virus were used in the virus chimera. Please check throughout the manuscript if it is made clear how a chimeric virus is built up.

115)   Line 200 and below: The statement: ‘Vector with the FAP-BiTE transgene was cytotoxic for normal human fibroblasts in the presence of T cells after infecting the ovarian cancer cells (SCOV-3). In contrast, the non-transformed vector was not toxic for those cells.’ is unclear. Please provide more info and explain what is happening here.

116)   Line 208: It is not clear what the authors intend to tell with the statement: ‘However, in some cases, a transgene can reduce the application of the vector.’

117)   Also the statement at line 235: ‘Results indicated that ColoAd1 had 50-fold more significant oncolytic activity than AdV-C5.’, is unclear please rephrase.

118)   Line 265: what is intended with the phrase: ‘However, species D adenoviruses are less immunogenic than species C, which limits the vectors' transduction and efficiency [9,30].’ It is difficult to envision how a reduced immunogenicity could limit the virus transduction. Is this really what the authors intended to state here. If so, please explain.

119)   Line 285: please explain what is so informative in the Weaver data.

220)   Line 309: ‘This serotype and, at the same time, the whole species were most likely formed due to the merger of human species B adenovirus with a chimpanzee adenovirus ChAdOx1 (also named Y25) [9,51].’ This is a statement that needs explanation, as it is unclear what the species B has to do with the jump from an HAdV-E from non-human primates to humans? Ref 9 and 51 do not refer to primary data on this virus.

221)   Line 329: Please explain why it is relevant that the long fiber of HAdV-F40 acts as a strong splice acceptor.  

222)   Line 333: HAdV-F viruses grow to very low titers in cultured human cells. This makes it rather difficult to work with these viruses. Hence they were indicated with the ‘fastidious adenoviruses’.  This is a more faithful reason.

223)   Line 336: HAdV-G is not unique when HAdV-F also has two fibers. Please rephrase.

224)   Line 357: The fact that there are few HAdV-A serotypes is not the reason to neglect them (there are fewer serotypes in HAdV-E, F, and G). It is their propensity to induce cancers in rodents that led to their disregard.

225)   In chapter 5 it would be very useful to stress the risks of including immune modulating transgenes in replicating oncolytic vectors, especially if there is side effects upon overdosing the cytokine. This there is a realistic hazard in having Inf-beta or IL-2 encoded by a vector that can replicate in a patient.  

226)   Line 602: What is the ‘RGKD integrin’. Is the integrin-binding RGD domain intended?

227)   Line 614: please amend ‘impact’ to read ‘impacts’.

Author Response

(The authors gave the same response as above.)

Round 2

Reviewer 3 Report

The modifications improve the quality of the contents of the manuscript. In the current form the manuscript is acceptable. I have no further options.